# Greenhouse Gases Detection Exploiting a Multi-Wavelength Interband Cascade Laser Source in a Quartz-Enhanced Photoacoustic Sensor

**DOI:** 10.3390/s25082442

**Published:** 2025-04-12

**Authors:** Raffaele De Palo, Nicoletta Ardito, Andrea Zifarelli, Angelo Sampaolo, Marilena Giglio, Pietro Patimisco, Ezio Ranieri, Robert Weih, Josephine Nauschütz, Oliver König, Vincenzo Spagnolo

**Affiliations:** 1PolySense Lab, Dipartimento Interateneo di Fisica, University and Polytechnic of Bari, Via Amendola 173, 70126 Bari, Italy; raffaele.depalo@uniba.it (R.D.P.); n.ardito1@phd.poliba.it (N.A.); andrea.zifarelli@uniba.it (A.Z.); angelo.sampaolo@poliba.it (A.S.); pietro.patimisco@uniba.it (P.P.); vincenzoluigi.spagnolo@poliba.it (V.S.); 2PolySense Innovations srl, Via Amendola 173, 70126 Bari, Italy; 3Dipartimento di Biologia, Università degli Studi di Bari, Via Orabona 4, 70126 Bari, Italy; ezio.ranieri@uniba.it; 4Nanoplus Advanced Photonics Gerbrunn GmbH, Oberer Kirschberg 4, 97218 Gerbrunn, Germany; robert.weih@nanoplus.com (R.W.); josephine.nauschuetz@nanoplus.com (J.N.); oliver.koenig@nanoplus.com (O.K.)

**Keywords:** quartz-enhanced photoacoustic spectroscopy, interband cascade lasers, greenhouse gases

## Abstract

This study presents the performance of a multi-gas sensor for greenhouse detection based on quartz-enhanced photoacoustic spectroscopy (QEPAS). The QEPAS sensor exploits an innovative, compact three-wavelength laser module as excitation source. The module integrates three interband cascade laser chips with a beam combining system, all enclosed in a compact metallic package with sizes of 40 × 52 × 17 mm to generate a single output beam. The multi-gas QEPAS sensor was tested in a laboratory environment for the sequential detection of two greenhouse gases, methane (CH_4_) and carbon dioxide (CO_2_), and a precursor greenhouse gas, carbon monoxide (CO). At an integration time of 100 ms, minimum detection limits of 21 ppb, 363 ppb, and 156 ppb, were estimated for CH_4_, CO_2,_ and CO detection, respectively, all well below their natural abundance in air.

## 1. Introduction

In 2023, global temperatures reached unprecedented levels, marking it as the warmest year on record with an increase of 1.45 °C above pre-industrial levels. The World Meteorological Organization (WMO) has highlighted concerning trends in key climate indicators, including greenhouse gas (GHG) concentrations and global temperatures, emphasizing the critical role of methane (CH_4_) and carbon dioxide (CO_2_) in driving global warming [1,2].

Carbon dioxide is the most prevalent anthropogenic GHG, primarily arising from fossil fuel combustion, vehicle exhaust, industrial processes, and deforestation [2]. Recent data from NOAA’s Global Monitoring Laboratory revealed that in 2023, the global average atmospheric CO_2_ concentration reached a new record of 419.3 parts per million (ppm), an increase of 2.8 ppm with respect to the previous year [3].

Methane, despite its low concentration (~1.8 ppm), exerts significant influence on global climate due to its heat-trapping capability. Over a 100-year period, it has a global warming potential (GWP) approximately 25 times greater than CO_2_ [2,4]. Beyond its climatic effects, methane also contributes to local air quality degradation as a precursor to ground-level ozone (O_3_). Major sources of methane emissions include fossil fuel production (especially natural gas), vehicle exhaust, and agricultural activities [2,5].

Together with methane and CO_2_, carbon monoxide (CO) also acts as a precursor GHG influencing atmospheric chemistry and human health [6]. Originating from incomplete combustion in industrial and vehicular activities, CO interacts with hydroxyl radicals (OH) in the atmosphere to produce CO_2_, thereby increasing its concentration in the atmosphere. Additionally, CO affects methane concentrations in the atmosphere by competing with CH_4_ for OH. The oxidation of methane primarily occurs through its reaction with OH, leading to the formation of water (H_2_O) and CO_2_. This oxidation process is crucial for reducing methane concentrations in the atmosphere and mitigating its overall impact as a greenhouse gas. When CO is present in elevated concentrations, it can significantly decrease the availability of OH, thereby influencing both the atmospheric lifetime and concentration of methane. Moreover, CO poses significant health risks even at low levels because it reduces the ability of hemoglobin to transport oxygen in the blood. The World Health Organization (WHO) advises stringent limits for short-term exposure (25–35 ppm for one-hour exposure) to mitigate health impacts, highlighting the dual environmental and public health challenges [7].

The 2023 Annual Report from the United Nations Environment Programme (UNEP) underscores the urgent need for a global response to the growing challenges of climate change, loss of biodiversity, and pollution [8]. In this scenario, a strategic priority is the creation of cost-effective stations for continuous, accurate, and selective monitoring of greenhouse gases (GHGs) in the atmosphere, which are also crucial for evaluating the effectiveness of interventions in achieving the desired outcomes. To address this requirement, GHGs sensors are continuously developed and improved. Among these, electrochemical, semiconductor-based, and optical-based sensors have gained popularity due to their cost-effectiveness and their compact design. Electrochemical sensors, like the Alphasense CO-A4, achieve MDLs of a few ppm but face challenges such as short service life and poor selectivity due to cross-interference with other gases [9]. Semiconductor-based sensors, such as Figaro Inc.’s TGS 3870-B00, are remarkable for their compactness and affordability in detecting gases like carbon dioxide and methane [10]. However, their methane detection range, typically in the thousands of ppm, limits their suitability for environmental monitoring. NDIR sensors, like the Senseair K30, offer sensitivity ranging from low ppm to percentage levels, with power consumption typically between 50 mW and a few watts, depending on the gas type and configuration. On the other hand, they are not highly selective and suffer from spectral interference by water [11,12]. Asking for high sensitivity and specificity, laser-based gas sensing techniques occupy a dominant role in trace gas detection. These methods also fulfill the requirements for real-time monitoring thanks to their fast response time [13,14,15,16]. For instance, Cavity Ring-Down Spectroscopy (CRDS) sensors, such as the Picarro Gas Concentration Analyzer, offer sensitivity levels down to a few parts-per-billion (ppb) for methane and carbon monoxide [17,18]. Nevertheless, CRDS systems require perfect optical alignment, and their high cost, several kilo euros, along with a bulky form factor, limits their use in environmental monitoring applications. In this context, Quartz Enhanced Photoacoustic Spectroscopy (QEPAS) stands out as a reliable and robust technique for the detection of several trace gas species [19,20,21,22,23,24,25]. QEPAS exploits the principles of photoacoustic spectroscopy, employing a quartz tuning fork (QTF) as an acoustic transducer [26]. Photoacoustic effect occurs when periodically modulated light is absorbed by a gas sample. Through non-radiative relaxation pathways, the modulated absorption is converted into pressure waves, which can be detected using a transducer. In QEPAS, these waves are detected by a QTF and converted into an electrical signal through the piezoelectric effect occurring in the quartz crystal. To enhance the sensitivity of the QEPAS sensor, the QTF is typically coupled with a pair of millimeter-size resonator tubes to amplify the pressure waves. The QTF response is independent of the exciting light source wavelength, making it a promising candidate for use with multi-wavelength sources. Recently, a QEPAS sensor for multi-gas detection with a multi-wavelength source consisting of three commercial Quantum Cascade Lasers (QCLs) within a single box has been proposed [27]. Three QCLs housed in HHL packages were mounted into an aluminum box and manually aligned to be collinear, using free-space optics mounted on standard mechanical supports. The resulting assembly is bulky, uses a lot of energy, and faces several thermal management challenges related to the operation of each HHL package. Additionally, the stability of the beam collinearity is significantly affected by extra-thermal heating inside the enclosure, especially when three QCLs are turned on simultaneously. This suggests that, to achieve successful long-term stability in multi-laser sources, a different technological approach is needed: instead of assembling multiple laser sources in one box, it involves integrating various laser chips into a single package. This minimizes the system’s footprint, which is particularly important in space-constrained environments. Moreover, the optical assembly required for ensuring the collinearity of different beams must be integrated in a compact way, avoiding the use of large mechanical supports. The optical components should be securely fixed to minimize internal misalignments caused by thermal expansion. To improve thermal extraction, the use of low-power consumption laser chips becomes mandatory. Furthermore, from a market perspective, manufacturing and assembling multiple laser chips into a single module can reduce packaging costs, simplify the alignment process, and streamline production, leading to lower overall costs compared to using individual laser modules.

In this work, an innovative three-wavelength ICLs module was employed as a light source for a compact multi-gas QEPAS sensor [28,29,30,31]. The module offers remarkable compactness and ruggedness, being characterized by reduced dimensions (~5 cm on the longest side) and a robust design. It incorporates three distinct ICL chips, which are combined into a single output beam through a custom-designed integrated beam combining system made of lenses and dichroic mirrors. The 3λ-module was employed as laser source for a QEPAS sensor for the sequential detection of direct and precursor GHGs, namely CH_4_, CO_2_, and CO. Using this configuration, a laboratory benchtop sensor prototype was developed, and the three gases were targeted in a controlled environment as proof of concept.

## 2. Materials and Methods

### 2.1. Three-Wavelength ICL Module

The laser source is a multi-wavelength module, realized by Nanoplus Advanced Photonics Gerbrunn GmbH, consisting of three ICL chips with central emission wavelengths of 3.35 µm, 4.23 µm, and 4.57 µm, selected for detecting CH_4,_ CO_2,_ and CO, respectively. The laser chips are mounted on aluminum nitride heat spreaders and arranged on sub-mounts within aluminum housing, which includes Peltier elements for the thermal management of each laser. The power consumption of each ICL is lower than 0.7 W. A schematic illustration of the module is depicted in Figure 1a.

The module measures 40 × 52 × 17 mm and integrates a beam combining system consisting of three lenses (L1, L2, and L3) and three dichroic mirrors (M1, M2, and M3), designed to obtain a single output beam. Each ICL beam is focused using a dedicated lens that is mounted on the same sub-mount as the laser chip, ensuring stable thermal performance. The beams of the different lasers are focused onto the focal plane of the ADM. This is achieved by first focusing each laser beam to the corresponding path length using a lens and then combining the beams using dichroic mirrors. A high-resolution pyrocamera (Pyrocam III, Ophir Spiricon PY-III-C-C) with pixel size of 0.1×0.1 mm was placed at the focal plane of the 3λ-module to analyze the overlap of three ICL beams. The resulting intensity distribution of the three ICL beams is shown in Figure 1b. The overall light distribution diameters in both the x- and y-directions were measured to be 0.8 mm.

### 2.2. Experimental Setup for Multi-Gas QEPAS Sensor

The 3λ-module was used as a light source in a QEPAS sensor setup schematically illustrated in Figure 2.

The Acoustic Detection Module, ADM01, provided by Thorlabs GmbH, was equipped with two windows (WG70530-E4) with AR coating in the 2–5 µm range. ADM01 contains a spectrophone, which consists of a T-shaped QTF acoustically coupled with a pair of millimeter-size resonator tubes arranged on both sides of the QTF, in an on-beam configuration. Each tube has a length of 12.4 mm, and an internal and external diameter of 1.59 and 1.83 mm, respectively [32]. The transmitted optical power through the ADM was monitored by a power meter to refine the optical alignment. The gas handling system included a gas mixer (MCQ Instruments, Gas Blender 103), mounted upstream to maintain a gas flow rate of 50 sccm; downstream, a pressure controller (Alicat, MC3S-200SCCM) and a vacuum pump fixed the operating pressure at 400 Torr. In these operating conditions, the spectrophone had a resonance frequency of $f_{0}=12,439.4$ Hz with a quality factor of 14,650. The 3λ-module QEPAS sensor operated in wavelength modulation with dual-frequency detection (2f-WM) [33]. A sinusoidal dither at half of the resonance frequency of the spectrophone, $f=f_{0}/2$, was applied to the ICL current driver (ITC4002QCL, Benchtop Laser Driver and Temperature Controller, Thorlabs) together with a low-frequency ramp to scan across the absorption feature. The resulting QTF signal was demodulated at $f_{0}$ by a lock-in amplifier (Zurich Instruments MFI 500 kHz Lock-in Amplifier) with an integration time of 0.1 s and a roll-off of 12 dB/oct.

## 3. Results

This section presents the results obtained using the 3λ-module QEPAS sensor for the in-sequence detection of CH_4_, CO_2,_ and CO. The performance of the sensor was evaluated by operating the three ICL chips individually: the 3.35 µm laser chip for the CH_4_ detection, the 4.23 µm chip for CO_2_ detection and the 4.57 µm chip for CO detection. The temperature of the module was maintained at 15 °C using a temperature controller to regulate the Peltier cell. For each analyte, the sensor calibration was performed by diluting a certified mixture in nitrogen (N_2_) with pure N_2_ and acquiring QEPAS spectral scans at different concentrations across the selected absorption feature.

### 3.1. Direct Greenhouse Gases Detection

#### 3.1.1. Methane Detection

Methane was the first analyte investigated. Its absorption cross-section was reconstructed using the HITRAN database [34], simulating at 400 Torr a mixture of 1.8 ppm of CH_4_ in N_2_ within the 2988.4–2989.4 cm^−1^ range, which corresponds to the spectral dynamic range of the 3.35 µm-chip. In addition, the absorption cross-section of mixtures of 420 ppm of CO_2_ in N_2_, 100 ppb of CO in N_2,_ and 0.4% of H_2_O in N_2_ were also included in the simulation to evaluate potential spectral interferences with CH_4_ detection. The simulated analyte concentrations are equal to the estimated atmospheric abundance of the target species [3,7,33]. The results are shown in Figure 3.

The simulation reveals an absorption feature of water vapor at 2988.62 cm^−1^, which is sufficiently distant from the CH_4_ triplet, characterized by three distinct peaks at 2988.82 cm^−1^, 2988.98 cm^−1^, and 2989.08 cm^−1^, all with similar cross-sections.

The response of the 3λ-ICL QEPAS sensor for CH_4_ detection was evaluated by turning on the 3.35 µm chip. Starting from a certified concentration of 50 ppm of CH_4_ in N_2_, different mixtures were generated with methane concentrations in the range 10–50 ppm in N_2_. Figure 4a reports the CH_4_ QEPAS spectral scans acquired in the full spectral range of the 3.35 µm chip at different methane concentrations.

The acquired QEPAS spectral scan matches the CH_4_ simulation in Figure 3 with the strongest absorption feature observed at an ICL current of 97.5 mA, corresponding to a laser emission at 2988.82 cm^−1^, with an optical power of 20 mW. For each spectral scan, the peak value of this intense feature was extracted and plotted as a function of the CH_4_ concentration, as shown in Figure 4b. A linear fit of the experimental data points was performed to obtain the sensor calibration curve for methane, depicted as a red solid line in Figure 4b, with an R-squared value higher than 0.999. The sensor sensitivity was estimated from the slope of the best fit, resulting in 7.70 ± 0.11 mV/ppm. A 1-σ noise level of 0.16 mV was measured by flowing N_2_ through the ADM, while the laser current was fixed at 97.5 mA. The minimum detection limit (MDL) was calculated as the concentration corresponding to a signal-to-noise ratio of 1. For CH_4_, an MDL of 21 ppb was estimated with a 0.1 s lock-in integration time, well below its natural atmospheric concentration.

#### 3.1.2. Carbon Dioxide Detection

The absorption cross-section of carbon dioxide was reconstructed using the HITRAN database for a mixture of 420 ppm of CO_2_ in N_2_ within the spectral range of 4.23 µm chip, from 2363.3 to 2364.8 cm^−1^, at 400 Torr. As before, the results of simulation are reported in Figure 5 together with the simulations of a mixture of 1.8 ppm of CH_4_ in N_2_, a mixture of 100 ppb CO in N_2,_ and a mixture of 0.4% H_2_O in N_2_, in the same spectral range.

A well-isolated CO_2_ absorption feature located at 2364.1 cm^−1^ can be detected by the 4.23 µm chip at an injection current of 125.8 mA and with an optical power of 1.4 mW, completely free from interference by the other two gases. The CO_2_ 2f-spectral scans were measured with the 4.23 µm ICL chip with different mixtures of CO_2_ in the range of 250–1000 ppm in N_2_. The acquired CO_2_ 2f-spectral scans are shown in Figure 6a.

For each spectral scan, the peak values were extracted and plotted as a function of the CO_2_ concentrations in Figure 6b. The linear fit yields a detection sensitivity of 0.44 ± 0.01 mV/ppm based on the linear regression of data points. With a measured 1-σ noise level of 0.16 mV, estimated as previously described, and an MDL of 363 ppb for CO_2_ detection, well below its natural concentration, was estimated at 0.1 s lock-in integration time.

### 3.2. Precursor Greenhouse Gas Detection: Carbon Monoxide

The QEPAS sensor calibration for carbon monoxide was performed employing the same method and procedure as the direct greenhouse gases. The absorption cross-section of carbon monoxide was simulated with the HITRAN database within the spectral emission of the 4.57 µm chip, from 2189.6 to 2190.4 cm^−1^. To evaluate potential spectral interferences, a mixture of 100 ppb of CO in N_2_, a mixture of 1.8 ppm of CH_4_ in N_2_, a mixture of 420 ppm of CO_2_ in N_2,_ and a mixture of 0.4% H_2_O in N_2_ were also included in the simulation reported in Figure 7.

The isolated CO absorption feature located at 2190 cm^−1^, free from interference by the other analytes, has been reconstructed using the 4.57 µm ICL chip at an injection current of 144.7 mA with an optical power of 8.4 mW. Different 2f-spectral scans of CO measured at different CO concentrations in N_2_ are plotted in Figure 8a.

The calibration curve was obtained by plotting the peak values of the spectral scans as a function of the CO concentration, as shown in Figure 8b. The error bars represent the standard deviation of the QEPAS signal, evaluated as previously described. From the best linear fit, a sensitivity of 1.15 ± 0.05 mV/ppm was determined. With a 1-σ noise level of 0.17 mV, the MDL was estimated to be 148 ppb at a lock-in integration time of 0.1 s, which is below the WHO one-hour average safe limit.

### 3.3. Long-Term Stability of the Sensor

To assess the stability of the sensor and estimate the 1-σ noise as a function of the lock-in integration time, an Allan–Werle deviation analysis of the 3λ-ICL QEPAS sensor was performed. This analysis involved simultaneously activating all three ICL chips at their respective current values, which correspond to the peak values extracted from spectral scans for calibration. Measurements were taken over a seven-hour acquisition period at 0.1 s lock-in integration time, with N_2_ flowing in the ADM01 at 400 Torr and a gas flow rate of 50 sccm. The results of the Allan–Werle analysis are presented in Figure 9.

The noise level decreases with the integration time following the expected trend of ~1/√t. This indicates the predominance of QTF thermal noise as the noise source. For integration times exceeding 600 s, the noise performance begins to degrade due to slow mechanical vibrations, which become increasingly relevant over longer periods. With a lock-in integration time of 10 s, the 1-σ noise of the 3λ-ICL QEPAS signal can be reduced to 0.06 mV, thus achieving an MDL of 8 ppb, 136 ppb, and 55 ppb for CH_4_, CO_2,_ and CO detection, respectively.

### 3.4. Gas Mixtures Detection and Analysis

The multi-gas detection capabilities of the QEPAS sensor were further validated by performing measurements on three different gaseous samples which were synthetically generated in laboratory environment with the following mixing ratios:Mix #1: 500 ppm CO_2_, 12.5 ppm CH_4_, 250 ppm CO, in N_2_;Mix #2: 250 ppm CO_2_, 25 ppm CH_4_, 250 ppm CO, in N_2_;Mix #3: 250 ppm CO_2_, 12.5 ppm CH_4_, 500 ppm CO, in N_2_.

The 2f-spectral scans of the analytes were obtained by sequentially activating the three laser chips, with the sensor operating under the same experimental conditions employed for the calibration phase. Each scan was acquired with a lock-in integration time of 100 ms, and the results are shown in Figure 10.

The QEPAS signals obtained for each analyte in the multi-gas mixtures closely match the corresponding single-gas signals measured during the sensor calibration (Figure 4, Figure 6 and Figure 8). No significant signal distortion or interference was detected, confirming that the presence of additional gas components in the mixtures had no impact on the spectral response.

For each spectral scan, the peak value was extracted and converted into a concentration using the calibration curves derived in the previous section. Table 1 summarizes the expected and estimated concentrations for each analyte, along with the associated signals and the relative differences between expected and estimated values.

The errors associated with the expected concentrations were calculated by combining the expanded uncertainties of the certified gas cylinders concentrations provided by the manufacturer (4% at 3-σ) and the gas mixer setpoint accuracy (1% at 1-σ). Signal fluctuation was determined as the 1-σ standard deviation from a 10 min acquisition while flushing the gas mixture through the sensor. This value was incorporated into the error propagation, along with the sensitivity uncertainty, to compute the total error in the estimated concentrations. For all mixtures, the analytes expected and estimated concentrations are comparable within the relative uncertainty intervals, demonstrating the sensor’s reliability.

## 4. Conclusions

In this work, a novel three-wavelength ICL module was employed as a light source for a multi-gas QEPAS sensor. The module integrates three chips with emission wavelengths of 3.35 µm, 4.23 µm, and 4.57 µm, properly selected for detecting CH_4,_ CO_2_ and CO, respectively. The three beams were combined to generate a single focused output using an integrated beam combining system housed within a compact module, with overall sizes of 40 × 52 × 17 mm. The QEPAS sensor was calibrated in a laboratory environment for sequential detection of CH_4_, CO_2,_ and CO, achieving a sensitivity of 7.70 mV/ppm, 0.44 mV/ppm, and 1.09 mV/ppm, respectively. The measured 1-σ noise levels were 0.160 mV for CH_4_, 0.16 mV for CO_2_, and 0.17 mV for CO detection, resulting in estimated MDLs of 21 ppb, 363 ppb, and 148 ppb, respectively, at an integration time of 0.1 s, which, respectively, reduces to 8 ppb, 136 ppb, and 55 ppb for 10 s integration time.

To assess the impact of drift in the quartz tuning fork’s parameters on the long-term stability of the QEPAS sensor, its resonance frequency and quality factor were periodically monitored throughout the experiment. No significant variations were observed, confirming that drift had a negligible effect on the overall sensor performance.

The compact design of the 3λ-ICL module ensures reliable performance in a small form factor, making it an ideal source for a QEPAS sensor for real-time and multi-gas detection of GHGs and their GHG precursor CO. Considering the characteristics of the developed multi-gas sensor (compactness, versatility, detection limits, etc.), it represents a promising solution for urban air quality monitoring. Offering a cost-effective solution at a few thousand euros, it addresses a critical gap in gas detection technologies for long-term monitoring, with potential for significant cost reductions through large-scale production of the custom laser source. Finally, its efficient thermal management and compact size facilitate integration into mobile platforms, such as drones.

## Figures and Tables

**Figure 1 sensors-25-02442-f001:**
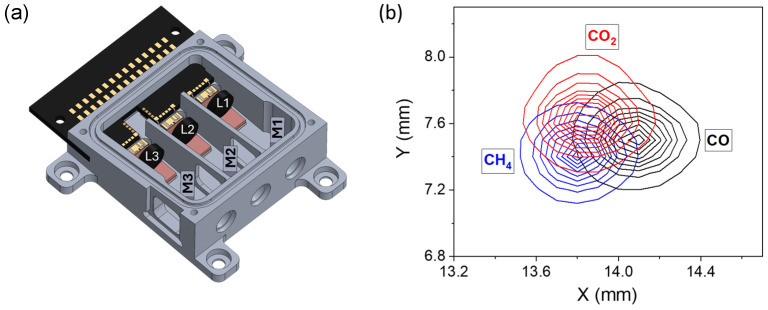
(**a**) Schematic of the three wavelength devices; L1, L2, L3, and L4 are lenses; M1 is a CaF_2_-substrate based mirror, while M2 and M3 are YAG-substrate based mirrors. (**b**) Beam profiles of the laser targeting CH_4_ (blue), CO_2_ (red), and CO (green) overlapped in the focal plane of the 3λ-module.

**Figure 2 sensors-25-02442-f002:**
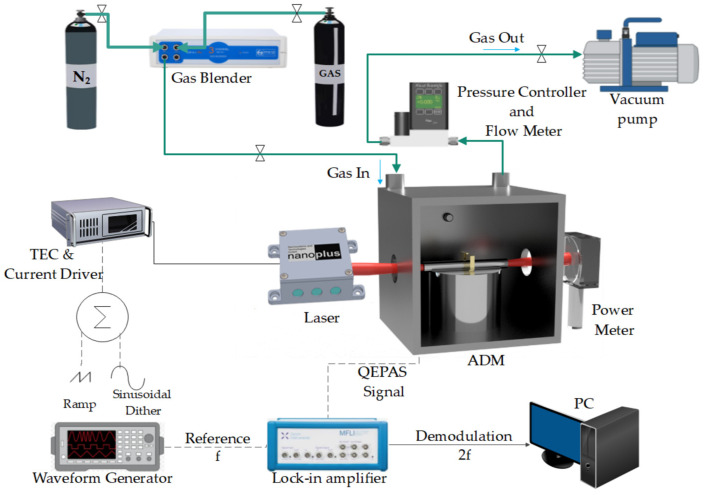
Schematic of the employed 3λ-ICL QEPAS setup; ADM, acoustic detection module; PC, personal computer.

**Figure 3 sensors-25-02442-f003:**
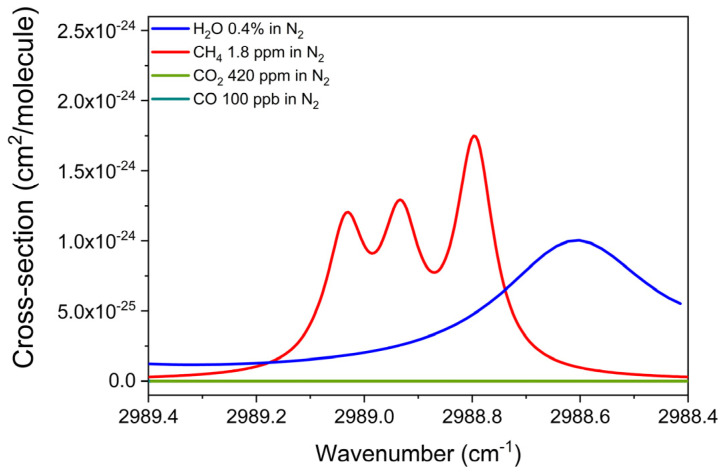
Absorption cross-section of a mixture of 1.8 ppm of CH_4_ in N_2_, a mixture of 420 ppm of CO_2_ in N_2_, a mixture of 100 ppb of CO in N_2,_ and a mixture of 0.4% of H_2_O in standard air, simulated with the HITRAN database within the emission spectral range of the 3.35 µm ICL at 400 Torr and room temperature.

**Figure 4 sensors-25-02442-f004:**
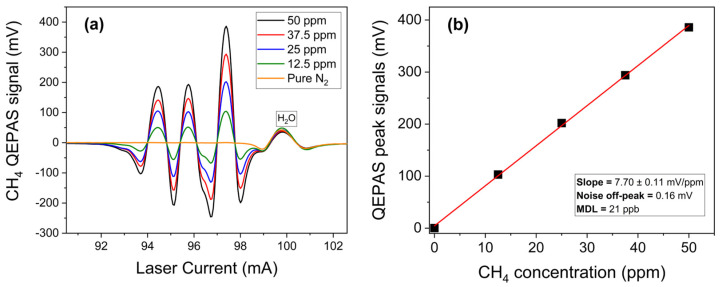
(**a**) QEPAS spectral scans measured for different concentrations of CH_4_ in N_2_ as a function of ICL injection current; (**b**) QEPAS signal of the most intense CH_4_ feature as a function of gas concentration (black squares) and the corresponding best linear fit (red solid line).

**Figure 5 sensors-25-02442-f005:**
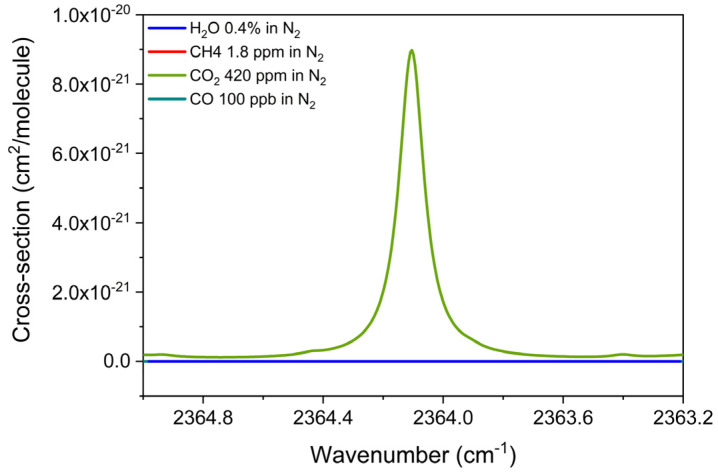
Absorption cross-section of a mixture of 420 ppm of CO_2_ in N_2,_ a mixture of 1.8 ppm of CH_4_ in N_2_, a mixture of 100 ppb of CO in N_2,_ and a mixture of 0.4% of H_2_O in standard air, simulated with the HITRAN database at 400 Torr and room temperature.

**Figure 6 sensors-25-02442-f006:**
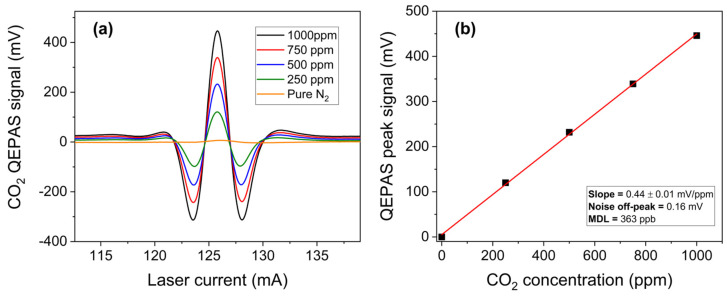
(**a**) QEPAS spectral scans measured for different concentration of CO_2_ in N_2_ and pure N_2_; (**b**) QEPAS signal of the absorption feature, targeted at an injection current of 125.8 mA for the 4.23 µm ICL, as a function of the CO_2_ concentration (black squares) with the corresponding best linear fit (red solid line).

**Figure 7 sensors-25-02442-f007:**
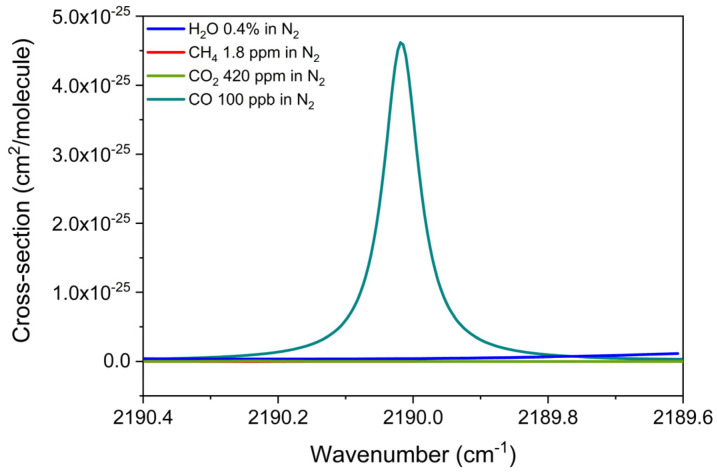
Absorption cross-section of a mixture of 100 ppb of CO in N_2_, a mixture of 1.8 ppm of CH_4_ in N_2_, a mixture of 420 ppm of CO_2_ in N_2_, and a mixture of 0.4% of H_2_O in standard air, simulated with the HITRAN database at 400 Torr and room temperature, within the 4.56 µm ICL spectral range.

**Figure 8 sensors-25-02442-f008:**
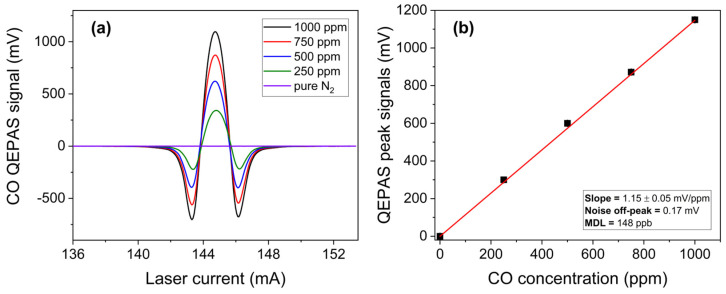
(**a**) QEPAS spectral scans measured for different concentrations of CO in N_2_ and pure N_2_; (**b**) QEPAS signal of the most intense feature as a function of the CO concentration (black squares) and the corresponding best linear fit (red solid line).

**Figure 9 sensors-25-02442-f009:**
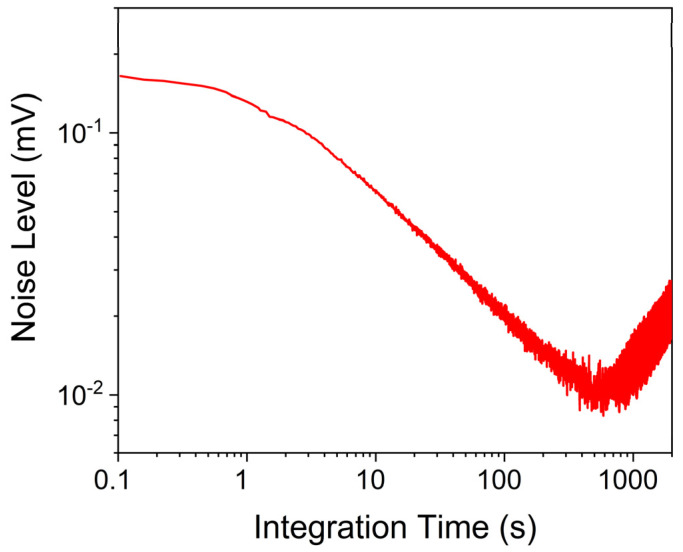
Allan–Werle deviation for the 3λ-ICL QEPAS noise level as a function of the integration time.

**Figure 10 sensors-25-02442-f010:**
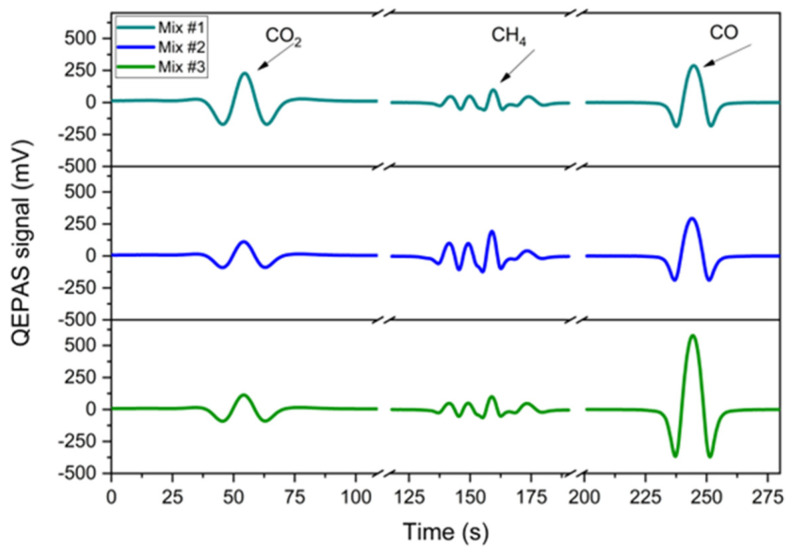
2f-QEPAS Spectral Scans of CO_2_, CH_4_, and CO in Gas Mixtures #1, #2, and #3. From left to right: spectral responses for CO_2_, CH_4_, and CO. The breakpoints in the spectra represent the time interval of approximately 10 s required for sequential detection between each laser.

**Table 1 sensors-25-02442-t001:** Expected and estimated concentrations for each analyte in the three mixtures, with associated signals and relative difference.

Mix	Target Gas	Extracted Signal (mV)	Expected Signal (mV)	Expected Concentration (ppm)	Estimated Concentration (ppm)	Difference (ppm)	Relative Difference (%)
#1	CO_2_	228.14 ± 0.17	220.0	500 ± 10	518.5 ± 11.8	18.5	3.7
#1	CH_4_	98.33 ± 0.16	96.3	12.5 ± 0.3	12.8 ± 0.2	0.3	2.2
#1	CO	288.14 ± 0.17	287.5	250 ± 5	250.9 ± 10.9	0.9	0.4
#2	CO_2_	111.17 ± 0.17	110.0	250 ± 5	255.3 ± 11.1	5.3	2.1
#2	CH_4_	99.84 ± 0.16	96.3	25 ± 0.5	24.9 ± 0.4	0.1	0.2
#2	CO	293.57 ± 0.17	287.5	250 ± 5	255.3 ± 11.1	5.3	2.1
#3	CO_2_	113.06 ± 0.17	110.0	250 ± 5.0	256.9 ± 5.9	6.9	2.7
#3	CH_4_	99.67 ± 0.16	96.3	12.5 ± 0.3	12.9 ± 0.2	0.4	3.6
#3	CO	578.78 ± 0.17	575	500 ± 10	503.3 ± 21.9	3.3	0.7

## Data Availability

Data are available on demand.

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
