# Peer review of "Greenhouse Gases Detection Exploiting a Multi-Wavelength Interband Cascade Laser Source in a Quartz-Enhanced Photoacoustic Sensor"

_sensors, 2025, doi:10.3390/s25082442_

Round 1
Reviewer 1 Report
Comments and Suggestions for Authors
please see attached PDF file

Author Response
Dear Reviewer,
please find in attachment our reply to your kind edits and notes.
Best regards

Reviewer 2 Report
Comments and Suggestions for Authors
This study presents the performance of a multi-gas sensor for greenhouse gases detection based on quartz-enhanced photoacoustic spectroscopy. The measured MDLs for CH4, CO2, and CO reduced to 8 ppb, 136 ppb and 58 ppb for 10 s integration time. This study could provide a new technical approach for greenhouse gas measurements and can be accepted with minor revision. However, the paper still has to make the following revisions:
- Quartz tuning fork, as the critical sensor component, needs to be described in detail with respect to their characteristic parameters, such as dimensions and bandwidth.
- The characteristic parameter drift of the quartz tuning fork will affect the stability of QEPAS in long-term measurements. Therefore, an outlook on the method of suppressing the characteristic parameter drift of the quartz tuning fork should be added in the Conclusion section.
- Figure 2 requires the employment of a higher resolution image. The sizes of two images in Figure 4 need to be resized.
- The citation format of references needs to be carefully checked.
- Some important QEPAS research advancements are missing from the introduction and need to be cited: In Situ High-Precision Measurement of Deep-Sea Dissolved Methane by Quartz-Enhanced Photoacoustic and Light-Induced Thermoelastic Spectroscopy. Analytical Chemistry.
Author Response

(The authors gave the same response as above.)

Reviewer 3 Report
Comments and Suggestions for Authors
Questions:
The article does not mention the reason for choosing the total pressure of the gas mixture of 400 Torr. In (Adv. Photonics Res. 2023, DOI: 10.1002/adpr.202200353) I found an explanation, but this system (3λ laser module + ADM) is supposed to be used for atmospheric sensing, then it would be advisable to carry out measurements and calibrate the photoacoustic sensor at an atmospheric gas of 750 Torr, since the spectral lines of the gas due to impact broadening of the article, and this will affect the recorded acoustic spectrum of the gas.
Notes:
There is an error in the diagram of the gas supply and storage system in the photoacoustic cell. The pressure regulator and flow meter are drawn strangely. The purpose of this regulator is to stabilize the pressure in the photoacoustic cell, and not in front of the cell. Such regulators have one input and one output. Perhaps it is worth using the drawing from previous publications, it is drawn more clearly there.
Author Response

(The authors gave the same response as above.)

Round 2
Reviewer 1 Report
Comments and Suggestions for Authors
I would like to thank the authors for addressing my comments and suggestions.